# Source of Dietary Fat in Pig Diet Affects Adipose Expression of Genes Related to Cancer, Cardiovascular, and Neurodegenerative Diseases

**DOI:** 10.3390/genes10120948

**Published:** 2019-11-20

**Authors:** Maria Oczkowicz, Tomasz Szmatoła, Małgorzata Świątkiewicz

**Affiliations:** 1Department of Animal Molecular Biology, National Research Institute of Animal Production, ul. Krakowska 1, 32-083 Balice, Poland; tomasz.szmatola@izoo.krakow.pl; 2University Centre of Veterinary Medicine, University of Agriculture in Kraków, Al. Mickiewicza 24/28, 30-059 Kraków, Poland; 3Department of Animal Nutrition and Feed Science. National Research Institute of Animal Production, ul. Krakowska 1, 32-083 Balice, Poland; m.swiatkiewicz@izoo.krakow.pl

**Keywords:** pigs, fatty acids, gene expression, *PLAU*, *CYBB*, *NCF2*, *ZNF127*

## Abstract

It has been known for many years that excessive consumption of saturated fats has proatherogenic properties, contrary to unsaturated fats. However, the molecular mechanism covering these effects is not fully understood. In this paper, we aimed to identify differentially expressed genes (DEGs) using RNA-sequencing, following feeding pigs with different sources of fat. After comparison of adipose samples from three dietary groups (rapeseed oil (*n* = 6), beef tallow (*n* = 5), coconut oil (*n* = 5)), we identified 29 DEGs (adjusted *p*-value < 0.05, fold change > 1.3) between beef tallow and rapeseed oil and 2 genes between coconut oil and rapeseed oil groups. No differentially expressed genes were observed between coconut oil and beef tallow groups. Almost all 29 DEGs between rapeseed oil and beef tallow groups are connected to neurodegenerative, cardiovascular diseases, or cancer (e.g., *PLAU*, *CYBB*, *NCF2*, *ZNF217*, *CHAC1*, *CTCFL*). Functional analysis of these genes revealed that they are associated with fluid shear stress response, complement and coagulation cascade, ROS signaling, neurogenesis, and regulation of protein binding and protein catabolic processes. Furthermore, gene set enrichment analysis (GSEA) of the whole datasets from all three comparisons suggests that both beef tallow and coconut oil may trigger changes in the expression level of genes crucial in the pathogenesis of civilization diseases.

## 1. Introduction

Inappropriate dietary patterns are one of the main risk factors for the development of cardiovascular disease (CVD). According to the Japanese Atherosclerosis Society (JAS), the recommended intake of fat should not exceed 20–25% of total energy intake, including 4.5–7% of saturated fatty acids [1]. The Mediterranean diet, rich in fish and monounsaturated fats, is the best example of diets for health promotion. Inhibiting the development of cardiovascular disease, neurodegenerative disease, and cancer by this diet has been well proved [2,3]. However, it is still unclear which components of the Mediterranean diet provide health-promoting agents. It is generally accepted that replacing some saturated fats (SFA) with unsaturated fats (UFA) may reduce the risk of heart and vascular disease, but it is not clear whether monounsaturated or polyunsaturated fats are more beneficial. Moreover, optimal fatty acid composition recommendations in the human diet are still unavailable [4]. 

There is currently a lack of complete knowledge of the molecular mechanisms responsible for observed health-promoting effects of specific fatty acids. Nutrigenomics provides us with the opportunity to gain insights into these mechanisms by observation of expression changes in genes associated with specific pathways and biological processes. In recent years, adipose tissue has been considered as a metabolically active organ that participates in the pathogenesis of metabolic diseases, by secreting various bioactive substances, such as proinflammatory cytokines. Moreover, obesity-induced, low-grade inflammation of adipose tissue is considered as a pathomechanism of CVD, increased thrombosis, diabetes, hyperlipidemia, and hypertension [5]. 

The pig was proposed as an animal model for atherogenesis almost twenty years ago [6]. Currently, pigs have become a promising alternative animal model to rodents due to many similarities in the size of organs and physiology with the human body. Moreover, pig breeding is aimed at increasing the high daily weight gain of animals, which makes it a natural model for an obesity-inducing lifestyle. In our experiment, we compared three dietary fats with different fatty acids composition. The selected fats: rapeseed oil—vegetable fat with a high content of UFA, beef tallow—animal fat with a high content of SFA (predominately long-chain SFA), and coconut oil—plant fat with a high content of SFA (predominately medium-chain SFA) are often used in the human diet. This study is performed on a previously published dataset of RNA-seq results (accession number: GSE101433). Previously, we have investigated an effect of cDDGS (corn dried distilled grains with solubles) on the adipose transcriptome of pigs fed with different amounts of cDDGS and sources of fat [7]. We observed that none of the dietary factors (cDDGS or fat) affected body weight or body composition. Contrary, the fatty acid composition of adipose tissue differed among dietary groups and was highly correlated with the fatty acid composition of diets [8]. Adipose tissue from animals consuming rapeseed oil contained significantly higher amounts of unsaturated fatty acids [7]. This prompted us to investigate the effect of the source of dietary fat on the subcutaneous adipose transcriptome of pigs deeply. We aimed to identify differentially expressed genes and get insights into biological processes and metabolic pathways that are changed after this intervention.

## 2. Materials and Methods 

### 2.1. Animals and Diets

All procedures included in this study relating to the use of live animals were in agreement with the local Ethics Committee for Experiments with Animals in Cracow (Resolution No. 912 dated 26 April 2012). 

In this manuscript, we describe the analysis of the effect of source of dietary fat on gene expression in adipose tissue of pigs, based on part of the data obtained in a previously described RNA sequencing experiment [7]. All procedures describing animal housing are described elsewhere [8]. In brief, crossbred fatteners were kept in individual straw-bedded pens in uniform conditions. The animals were healthy, and the most possibly uniform as regards body weight. The diets of all of the groups were isonitrogenous and isoenergetic and were formulated to cover the nutritional requirements of the pigs. The animals were divided into three dietary groups in which the diets differed among each other in terms of fodder fat: 3% rapeseed oil (group I), 3% beef tallow (group II), and 3% coconut oil (group III) (Figure 1). The ingredient composition and nutritive value of the diets, as well as the fatty acid compositions of the fat sources and feed mixtures, are presented elsewhere [6]. Briefly, group I feed mixture contained 80% of UFA content (44% MUFA and 36% PUFA), group II contained 67% of UFA (32% MUFA and 35% PUFA), and group III contained 45% of UFA (16% MUFA and 29% PUFA). The ratio of UFA/SFA in the used feed mixtures was: 4.7, 2.1, and 0.8 for group I, II, and III, respectively. The experimental fattening lasted from 60 to 118 kg of live weight of the animals. At the end of the experiment, all the pigs were slaughtered by stunning with high-voltage electric tongs (voltage 240–400 V), and samples of subcutaneous adipose tissue from the area between the last thoracic and the first lumbar vertebrae were collected for transcriptome analysis. All samples were stored in a freezer (−85 °C) until analysis.

### 2.2. RNA Isolation, RNA-Seq, and qPCR Procedures

In this study, we used part of the data from GSE101433 (6 samples+ cDDGS+ rapeseed oil, 5 samples + c DDGS + beef tallow, and 5 samples + cDDGS + coconut oil) and reanalyzed them with the aim to investigate the effect of different sources of fat on the adipose transcriptome. The whole procedure of RNA isolation and RNA-seq has been published previously [7] for the whole dataset. Regarding this part of the experiment, total RNA was isolated from 16 samples (5–6 samples per group) using the Direct-zol RNA kit (Zymo Research, Irvine, CA, USA) according to the attached protocol. The quality and concentration of the obtained RNA were evaluated on a Tapestation Instrument (Agilent Technologies, Inc., Santa Clara, CA, USA). The libraries were constructed using The SMARTer Stranded Total RNA Sample Prep Kit-HI Mammalian (Takara, Clontech, Mountain View, CA, USA) to obtain strand-specific sequencing data. The entire protocol included several steps: ribosomal RNA depletion with RiboGone technology; first-strand cDNA synthesis and purification; Illumina-specific library amplification by PCR; and RNA-Seq library purification. The quality and concentration of the libraries were evaluated on a TapeStation 2000 Instrument (Agilent Technologies, Inc., Santa Clara, CA, USA) and Qubit fluorometer (Thermo Fisher Scientific, Waltham, MA, USA). RNA libraries with the best quality (*n* = 16 (5–6 per group)) were prepared for sequencing using standard Illumina protocols: the libraries were diluted to a final concentration of 10 nM, and 5 or 6 samples from all groups were pooled. Cluster generation was performed on a cBot Instrument (Illumina, Inc., San Diego, CA, USA) using the TruSeq SR Cluster Kit v3-cBot-HS. Sequencing (single-end) was performed in two replicates on an Illumina HiScanSQ 2000 in one flowcell with the TruSeq SBS Kit v3-HS (50-cycles). To avoid the batch effect, all samples were processed simultaneously during RNA isolation and library preparation. All samples were run on one flowcell for NGS sequencing. 

Validation of the RNA-seq results was performed for 6 genes (*CD200R1*, *CD209*, *F3*, *CYBB*, *ZNF217*, *PLAU*) by quantitative real-time PCR (qPCR). The same samples were used for RNA-seq and qPCR analysis. cDNA was synthesized from 500 ng of total RNA with the cDNA Archive Kit (Thermo Fisher Scientific, Waltham, MA). qPCR was performed in triplicates on a QuantStudio 7–Flex Instrument (Thermo Fisher Scientific, Waltham, MA, USA) under the fast thermal profile. The reaction mix contained 1 µL of cDNA, 5 µL of TaqMan™ Fast Advanced Master Mix (Thermo Fisher Scientific, Waltham, MA, USA), 3.23 µL of water, 0.17 µL of 60 × TaqMan assay for *OAZ1* (endogenous control) amplification (Assay ID: Ss03397505_u1), and 0.5 µL of 20 × TaqMan gene expression assay for amplification of the target gene (*CD200R1* Assay ID: Ss04324657_m1, *CD209* Assay ID: Ss03819234_g1, *F3* Assay ID: Ss03381417_u1, *CYBB* Assay ID: Ss03391378_m1, *ZNF217* Assay ID: Ss03390845_m1, and PLAU assay ID: Ss03391043_m1). The relative quantity (RQ) of each sample was calculated based on the ΔΔCt method using QuantStudio^TM^ 6 and 7 flex real-time PCR software. 

### 2.3. RNA-Seq Data Processing and Statistical Methods

Demultiplexing of the RNA-seq samples was performed with the bcl2fastq Conversion software v1.8.4 (Illumina). Next, the quality check, trimming of reads, and mapping of reads were conducted with FastQC 11.8, FLEXBAR 3.5.0, and TopHat 2.1.1 software, respectively. The mapping statistics and read counts were generated with samtools 1.9, RSeQC, and HTSeq-count 0.11.1 software, Gtf-Ensembl annotation 96. Differential expression analysis was performed using DEseq 2 software. Genes with *p*-adjusted < 0.05 (FDR—Benjamini–Hochberg (BH) adjustment) and fold-change > (1.3) were regarded as differentially expressed. Classification of differentially expressed genes (DEGs) and functional enrichment was performed with the DAVID functional annotation tool (https://david.ncifcrf.gov/summary.jsp) and STRING version 11.0 (https://string-db.org/cgi/input.pl? sessionId = OEsGZBwjRNJY). GSEA (gene set enrichment analysis) was performed using WebGestalt software (http://www.webgestalt.org/). To compare the RNA-seq results with the qPCR results, the Pearson correlation between the fold change obtained after RNA-seq and qPCR was calculated using SAS software. Statistical analysis of gene expression after the qPCR analysis was performed with the Mann–Whitney test.

## 3. Results

### 3.1. Identification of DEGs after RNA-Sequencing

The statistics of the RNA-sequencing are presented in detail elsewhere [7]. In brief, we obtained, on average, 15,534,039 filtered reads per sample. More than 82% of them were mapped to *Sus scrofa* 11.1. All gene expression data has been submitted previously to the GEO accession number (accession number: GSE101433). Using DESeq2 software, we performed comparisons of gene expression between all three dietary groups (rapeseed oil vs. beef tallow, rapeseed vs. coconut oil, beef tallow vs. coconut oil, the first group in the comparison is the reference group). We identified 29 DEGs in the rapeseed oil vs. beef tallow comparison (15 upregulated and 14 downregulated in the beef tallow group, and only two DEGs (*CD209* and *FAM16*) in the beef tallow vs. coconut oil comparison. No differentially expressed genes were observed in the rapeseed oil vs. coconut oil comparison (Table 1, Appendix A). Accordingly, only the first comparison—rapeseed oil vs. beef tallow—created hierarchical clusters (Figure 2).

Next, we performed a functional analysis of 29 DEGs identified in the rapeseed oil vs. beef tallow comparison with DAVID software (Appendix A). Despite the low number of genes, it revealed that the genes engaged in the hsa04610 KEGG pathway: complement and coagulation cascade were overrepresented (*p* < 0.004). Moreover, four REACTOME pathways (R-HSA-1236973: cross-presentation of particulate exogenous antigens (phagosomes) (Homo sapiens) *p* < 0.013; R-HSA-5668599: RHO GTPases activate NADPH oxidases (Homo sapiens) *p* < 0.021; R-HSA-1222556: ROS, RNS production in phagocytes (Homo sapiens) *p* < 0.054; and R-HSA-4420097: VEGFA-VEGFR2 pathway (Homo sapiens) *p* < 0.097 were enriched. Among biological processes, GO:0002479~antigen processing and presentation of exogenous peptide and presentation of exogenous peptide antigen via MHC class I, TAP-dependent, and GO:0045730~respiratory burst were the most significantly enriched processes. Consequently, genes engaged in GO:0043020~NADPH oxidase complex and GO:0016175~superoxide-generating NADPH oxidase activity were overrepresented among the cellular components and molecular functions. All the identified enrichments are presented in Figure 3 and Appendix A.

In addition, the functional analysis of genes identified in the rapeseed oil vs. beef tallow comparison (29 genes) by STRING software revealed that they create a significant gene network (PP enrichment = 0.023). Moreover, the genes engaged in fluid shear stress and complement and coagulation cascade were overrepresented (*p* < 0.00256, *p* < 0.0346) (Figure 4) after STRING analysis. 

Since the number of differentially expressed genes identified in our experiment was relatively small, we performed gene set enrichment analysis (GSEA) using all genes lists ranked by log2-fold change and *p*-value with WebGestalt software (http://www.webgestalt.org) (Figure 1). Figure 5 and Figure 6 present the enriched KEGG pathway (*Sus scrofa*) and Wikipathways (Homo sapiens) identified in this analysis for gene set from the rapeseed oil vs. beef tallow comparison. Enriched KEGG pathways and Wikipathways for other datasets (rapeseed oil vs. coconut oil and beef tallow vs. coconut oil) as well as enrichments for biological processes, cellular components, and molecular functions for all three comparisons are presented in Appendix A. 

In the dataset from rapeseed oil vs. beef tallow comparison, we observed several pathways connected to neurodegenerative diseases (e.g., Parkinson disease, Alzheimer disease), metabolic diseases (nonalcoholic fatty liver disease) and immunity (phagosome, the intestinal immune network for IgA production) (FDR < 0.05) (Figure 4). Moreover, several Wikipathways (Figure 6) connected to cancer (e.g., metabolic reprogramming in colon cancer, gastric cancer network I) and complement system (FDR < 0.05) were identified in this dataset. Additionally, we observed three enriched biological processes with FDR < 0.05 (generation of precursor metabolites and energy, endocrine process, purine-containing compound metabolic process) (Appendix A). Among cellular components, the respiratory chain with a positive enriched score (FDR < 0.05) as well as the extracellular matrix, sarcolemma, and collagen trimer with a negative enriched score (FDR < 0.05) were identified. Considering the rapeseed oil vs. coconut oil comparison dataset, we observed a high number of overlapping KEGG pathways and Wikipathways with the dataset from rapeseed oil vs. beef tallow comparison (e.g., oxidative phosphorylation, Parkinson disease, Alzheimer`s disease, metabolic reprogramming in colon cancer, TYROBP causal network). Moreover, the same biological processes with a positive score and FDR < 0.05 (generation of precursor metabolites and energy, endocrine process, purine-containing compound metabolic process) were present. Consequently, the same cellular components (respiratory chain, extracellular matrix, sarcolemma, and collagen trimmer) were enriched in these two datasets. 

Analysis of the dataset from the last comparison—beef tallow vs. coconut oil revealed high enrichment of pathways connected to immunity (e.g., intestinal immune network for IgA production, natural killer cells mediated cytotoxicity—KEGG pathways), (TYROBP causal network, Microglia pathogen phagocytosis pathway—Wikipathways) with positive enrichment score (FDR < 0.05) as well as enrichment of pathways connected to cellular respiration and mitochondria function (mitochondrial complex I assembly model OXPHOS system, oxidative phosphorylation—Wikipathways) with a negative enrichment score (FDR < 0.05).

### 3.2. Validation of RNA-Sequencing by qPCR

We selected six differentially expressed genes (*CD200R1*, *CD209*, *CYBB*, *ZNF217*, *F3*, *PLAU*) for qPCR analysis with TaqMan assays (Figure 7). Using this method, we observed the same direction of expression changes; moreover, the fold changes in gene expression between dietary groups obtained by RNA-seq and qPCR were highly correlated (R^2^ = 0.96, *p* < 0.0004). Scatter plots presenting the between RNA-seq and qPCR for each of the analyzed genes separately are presented in Appendix A. We also observed that the expression of *CD209* is significantly lower not only in the coconut oil but also in the beef tallow group compared to the rapeseed oil group, which was undetected by RNA-seq. Furthermore, the expression of the *PLAU* gene was significantly different not only in the rapeseed oil vs. beef tallow but also in the rapeseed oil vs. coconut oil and beef tallow vs. coconut oil comparisons (Figure 7). Two genes (*CD209* and *CD200R1*) were differentially expressed depending on sex 

## 4. Discussion

In this paper, we describe the impact of feeding various types of fat on changes in gene expression in the whole transcriptome scale. Our results suggest that these changes concern key genes involved in the pathogenesis of civilization diseases. The observed changes in expression are relatively small (31 DEGs, max fold change = 2.36). However, they are observed in animals in which no significant differences in phenotype were found, e.g., body weight, fat thickness, and differed only in the adipose fatty acids profiles. 

Both functional analysis software–STRING and DAVID–revealed that complement and coagulation cascade genes are overrepresented among DEGs identified in rapeseed oil vs. beef tallow comparison (Figure 3 and Appendix A). Three genes (*THBD*, *F3*, *PLAU*) are representatives of this KEGG pathway in our dataset. Additionally, the same genes belong to blood coagulation GO biological processes, which were also overrepresented. Differential expression of two genes (*PLAU*, *F3*) was confirmed by qPCR (Figure 4). According to qPCR, expression of *PLAU*-urokinase-type plasminogen activator–the gene engaged in the dissolution of fibrin by activating plasmin conversion from plasminogen—is 3.8 times higher in the rapeseed oil group than in the beef tallow group (Figure 4). The coconut oil group presented intermediate values statistically, different from the beef tallow group (Figure 4). Simultaneously, we observed overexpression of the *SRPX* gene in the rapeseed oil group. *SRPX* acts as a ligand for the urokinase plasminogen activator surface receptor and plays an important role in angiogenesis. On the contrary, *F3*–coagulation factor 3 expression was almost two times higher in the beef tallow vs. the rapeseed oil group, while the *THBD* (gene for thrombomodulin (an inhibitor of thrombine)) was downregulated in the beef tallow group. Additionally, expression of *MCFD2* (multiple coagulation factor deficiency 2)–the gene which facilitates the transport of coagulation factors V (FV) and VIII (FVIII)—was upregulated in the beef tallow group, indicating that higher production of FIII is accompanied by increased transport of FV and FVII. These results suggest that dietary beef tallow reduced fibrinolysis and induced coagulation in adipose tissue. It has been shown previously that fibrin and thrombin activity exacerbates diet-induced obesity, metabolic inflammation, and associated sequelae [9]. Interestingly, Lagrange [10] observed that linoleic and palmitic acids increased coagulation by enhancing thrombin generation in Zucker rat`s plasma. The diet based on beef tallow had the highest content of palmitic acid, which is in line with previous results. Interestingly, the diet based on the coconut oil diet had a similar amount of palmitic acid as the rapeseed oil diet. Simultaneously, *F3* and *THBD* gene expression were not statistically different from any diet in coconut oil, while expression of the *PLAU* gene displayed intermediate values, statistically different from the beef tallow group. This may suggest that not only palmitic acid but other fatty acids as well might modulate the expression of genes from complement and coagulation cascade.

GSEA, which included all genes ranked by log2fold change and *p*-value, partially confirmed the above-mentioned results. We observed that the dataset from rapeseed oil vs. beef tallow comparison was enriched for genes belonging to biological process coagulation (negative enriched score, FDR > 0.05); the same was observed for the dataset from rapeseed oil vs. coconut oil comparison. Additionally, the KEGG pathway: complement and coagulation cascade and Wikipathways: complement system and complement activation were enriched in this dataset (negative enrichment score, FDR < 0.05.). The negative enriched score observed in theses analysis suggest that most of the genes form theses pathways are downregulated in beef tallow and coconut oil when compared to rapeseed oil group. However, there are not only activators but also inhibitors of coagulation. Generally, results from GSEA imply the downregulation of complement and coagulation system processes in beef tallow and coconut oil, which is in contrast to results obtained after functional analysis based on DEGs only. 

The use of coconut oil in the human diet still remains controversial. Many opinions suggest that, despite containing high amounts of SFA, it is not as harmful as animal fats, although the American Heart Association presidential advisory advises against the use of coconut oil [11]. On the other hand, it has been found recently that coconut oil has a different effect to butter on blood lipids, being more comparable to olive oil with respect to increasing serum LDL-C [12]. Our results of DEGs add new information to this debate, indicating that the influence of coconut oil on fibrinolysis and coagulation in adipose tissue of pigs may be not as strong as that of beef tallow. However, results of GSEA (which is based on analysis of small changes, not necessarily statistically significant of the huge number of genes from one pathway) show that the relationship between dietary fats and coagulation may be more complicated. Taken together, we conclude from these results that the deleterious effect of dietary beef tallow on disorders connected to obesity may be triggered by impaired fibrinolysis (downregulation of the *PLAU* gene) and increased coagulation (increased expression of *F3* and decreased expression of *THBD*). However, we support the previously suggested need for the evaluation of the effects of individual fatty acids profile of fats on metabolism-related diseases [12].

The physiological role of the *PLAU* gene is not limited to blood coagulation but is much wider. Plasmin plays an important role in the degradation of betamyloid in the brain. Therefore, its activators and inhibitors have been connected to the pathogenesis of Alzheimer’s disease (AD). There have been several reports suggesting associations between polymorphisms in the PLAU gene and the incidence of Alzheimer’s disease. However, the results are inconsistent [13]. On the other hand, it was reported that knockout of the *PAI-1* gene (an inhibitor of plasminogen activators PLAU and tPA) significantly reduced the amounts of Aβ plaques in the brain of mice [14]. Moreover, the *PLAU* gene has been identified as related to Alzheimer’s disease via a network and pathway-based approach [15]. Our data suggest that the well-known effect of dietary n3-PUFA intake on the development of neurodegenerative diseases may be connected to improved beta amyloid degradation through the upregulation of expression of plasminogen activator (*PLAU*). However, this requires further study in the brain tissue. 

Functional analysis of genes differing in the level of expression after feeding a diet containing rapeseed oil vs. beef tallow showed that among the genes related to neurogenesis, protein catabolic processes and binding of proteins, e.g., (*ZNF217*, *CHAC1*) (*TIPARP*, *PLK2*, *PLXND1*) occur more often than expected. *ZNF217*, a member of the Krüppel-like family, is a transcription factor that functions as an oncogene in many cancer types [16]. In our study, we observed that dietary intake of beef tallow upregulates *ZNF217* expression 2-fold in relation to rapeseed oil, according to qPCR (Figure 7). Recently, it was found that *ZNF217* is overexpressed and enhances cell migration and invasion in colorectal carcinoma [17]. Beef tallow is rich in palmitic acid—a fatty acid with cancerogenic properties [18]. Increased expression of the *ZNF217* gene in the beef tallow dietary group may be one of the mechanisms explaining the high correlation between red meat consumption and increased occurrences of colorectal cancer [19] since red meat contains substantial amounts of palmitic acid. On the other hand, Wang [20] demonstrated recently that beta amyloid induced neurocytotoxity may be reduced by inhibiting *ZNF217* expression. Specifically, it was found that *ZNF217* is the target for mir-200, which reduces its expression. The authors concluded that the mir-200/*ZNF217* axis may be the potential therapeutic target in AD and cancer. Interestingly, GSEA revealed that genes responsible for metabolic reprogramming in colon cancer are overexpressed in beef tallow and coconut groups when compared to rapeseed oil group (Figure 6, Appendix A). Moreover, several pathways connected to neurodegenerative diseases (Alzheimer’s disease, Huntingtin disease, Parkinson’s disease) were enriched in beef tallow and coconut group when compared to rapeseed oil group (FDR < 0.05) (Figure 5 and Figure 6, Appendix A) One of the most important pathomechanisms of neurodegenerative diseases is mitochondria dysfunction. In line with this, we observed an enrichment of genes connected to mitochondria functions–mitophagy, oxidative phosphotylation, and TCA cycle, in beef tallow and in coconut oil groups. Considering GSEA results, we may speculate that beef tallow is more harmful in this respect than coconut oil since the Parkinson’s disease pathway and several other pathways connected to mitochondria function were enriched in beef tallow group when compared to both rapeseed oil and coconut oil (Appendix A). 

Another gene overexpressed by beef tallow diet with strong implications for human health is *CHAC1*; the gene codes for Glutathione-specific gamma-glutamylcyclotransferases 1, which catalyze the specific cleavage of glutathione (GSH) into 5-oxoproline (OPRO) and a cysteinylglycine (CysGly) dipeptide. GSH is one of the most potent natural antioxidants, widely distributed in almost all living organisms. It plays a pivotal role in the maintenance of redox homeostasis, making it an interesting target of drug therapies [21]. A decreased level of GSH accompanied by oxidative stress is observed in a great majority of diseases, including diabetes, cardiovascular diseases, Alzheimer’s disease, and HIV [22,23]. Reduced levels of GSH are also observed with aging [24]. The repletion of GSH has been proposed as a therapeutic strategy for these diseases. However, the delivery of GSH by oral administration is challenging [21]. Our results show that expression of *CHAC1* is affected by dietary fats since we observed increased *CHAC1* expression after beef tallow consumption compared to rapeseed oil. It suggests the opportunity to manipulate GSH level by diminishing its cleavage through decreased *CHAC1* expression by specific fatty acids. On the other hand, depletion of glutathione is considered as a strategy to increase the vulnerability of cancer cells to radiation and toxic drugs. It was observed that *CHAC1* was the most upregulated gene after Temozolomide therapy of glioma. Overexpression and knockdown of *CHAC1* expression significantly influenced TMZ-mediated cell viability, apoptosis, caspase-3 activation, and poly (ADP-ribose) polymerase (PARP) degradation [25]. Consequently, the possibility of increasing *CHAC1* expression by fatty acids has therapeutic potential as well. *PLK2* Polo-like kinase 2—another gene engaged in the degradation of proteins—was upregulated in the animals receiving rapeseed oil in the diet. Recently, a new degradation route of α-synuclein (a protein strongly linked to the pathogenesis of Parkinson’s disease) by *PLK2* has been described, offering new opportunities for the development of therapeutic strategies [26]. What is more, it has been shown that *PLK2* modulates α-synuclein protein levels by regulating its mRNA production as well [27].

Functional analysis with DAVID software revealed that genes engaged in the Reactome pathway; R-HSA-1222556: ROS, RNS production in phagocytes (Homo sapiens) are overrepresented among DEGs between rapeseed oil and beef tallow groups) (*p* < 0.055). The same was observed for GO Biological processes: GO:0045730~respiratory burst, GO:0042554~superoxide anion generation, GO:0006801~superoxide metabolic process, molecular function: GO:0016175~superoxide-generating NADPH oxidase activity, and cellular component: GO:0043020~NADPH oxidase complex. It shows that dietary fats affect the expression of key genes responsible for ROS production in phagocytes. According to RNA-seq, the expression of these genes *(CYBB* which encodes the heavy-chain component of phagocyte specific NADPH oxidase and *NCF2* which encodes neutrophil cytosolic factor 2, the 67-kilodalton cytosolic subunit of the multi-protein NADPH oxidase complex found in neutrophils) was approximately 80% higher in adipose of the animals receiving rapeseed oil.

At first glance, it is surprising since increased ROS production and oxidative stress are generally associated with arteriosclerosis and cardiovascular disease [28], and pharmacological inhibition of *CYBB* is associated with a delayed atherosclerotic progression in animal models [29]. Thus, we expected a higher expression of these genes in the animals exposed to beef tallow. Nevertheless, ROS production plays an important physiological role by signaling and enabling efficient pathogen and death cell cleaning by phagocytosis. What is more, pathological accumulation of diseased vascular cells and apoptotic cellular debris is observed in atherosclerosis [30]. Hence, we speculate that excessive consumption of animal fats may lead to impaired phagocytosis through decreased ROS production. Moreover, ROS production is connected to numerous modifications of cellular processes, including modifying the redox state of proteins and lipids. ROS functions are clearly concentration-dependent over a wide range of concentrations. However, due to the difficulty of reliable quantitative ROS detection, this issue remains not well understood and requires further investigations [31]. Interestingly, recently, it has been shown that cholesterol is a modulator of the phagocyte NADPH oxidase activity [32]. This is in accordance with our transcriptomic results since beef tallow is highly hypercholesterolemic [33]. Specifically, it was observed that cholesterol at high concentrations triggers inhibition of the activity of NADPH oxidase in the presence of pro-inflammatory signals [32]. 

Results of GSEA are in agreement with the above-mentioned information. We observed an enrichment of genes connected to phagosome (KEGG) and microglia pathogen phagocytosis pathway (Wikipathways) in beef tallow group when compared to rapeseed oil group (negative enrichment score, FDR < 0.05), which suggest a weakening of phagocytosis in adipose of animals receiving beef tallow in the diet. 

The effect of consuming different fats on the inflammatory state of adipose tissue is another interesting issue. It is generally accepted that during the progression of obesity infiltration of immune cells, increased M1/M2 macrophages ratio, impaired vascularization, and remodeling of the extracellular matrix occurs, which results in a low-grade inflammation of adipose tissue. In our experiment, we did not observe differences in fat weight, but we expected changes in the inflammation state of adipose since palmitic acid, present in beef tallow, is highly proinflammatory. Although we cannot clearly determine whether the fat consumed affected the inflammatory status of adipose tissue, we observed an increased expression of two anti-inflammatory markers (*CD209* and *CD200R1*) [34,35] in the fat of the animals receiving rapeseed oil in our experiment (Figure 4). Interestingly, the expression of these genes was higher in females than in the males), which supports recent observations that response to infectious diseases is gender-specific [36].

The results provided by GSEA regarding the effect of dietary fat on inflammatory processes and immune functions are ambiguous and difficult to interpret. Several KEGG pathways and Wikipathways were enriched with a negative enrichment score in rapeseed oil versus beef tallow comparison (e.g., natural killer T cell-mediated cytotoxicity, TYROBP causal network) suggesting that the number of immune cells of different classes were lower in beef tallow group. On the other hand, the adipocytokine signaling pathway was enriched with positive enrichment score in beef tallow when compared to rapeseed oil (Figure 5). Thus, we may speculate that after three months of dietary intervention, we observed the beginning of the immunological reorganization of adipose tissue in animals fed with beef tallow. 

Among the DEGs identified by us, there are several other genes related to the pathogenesis of civilization diseases. To mention only a few, we have observed different expression of the *CTCFL* gene—oncogene—which is tested as a target gene in the immunotherapy of cancer [37,38], *PADI2*—a novel angiogenesis-regulating gene, or *USP53*—which is a novel molecular biomarker for weight control [39].

## 5. Conclusions

In conclusion, our study suggests that dietary fats present in beef tallow and coconut oil may negatively affect human health by deregulation of expression of genes engaged in fibrinolysis, proteostasis, and proper function of phagocytes and mitochondria. These processes are important players in the pathogenesis of cardiovascular, neurodegenerative, and other metabolism-related diseases. Our observations provide support for the idea of using the pig as an animal model for human civilization diseases. These results, as a whole transcriptome study, support dozens of previously conducted experiments describing the effects of fatty acids on gene expression in vivo and in vitro. Nevertheless, there is still a need to prove these relationships at the protein and function levels.

## Figures and Tables

**Figure 1 genes-10-00948-f001:**
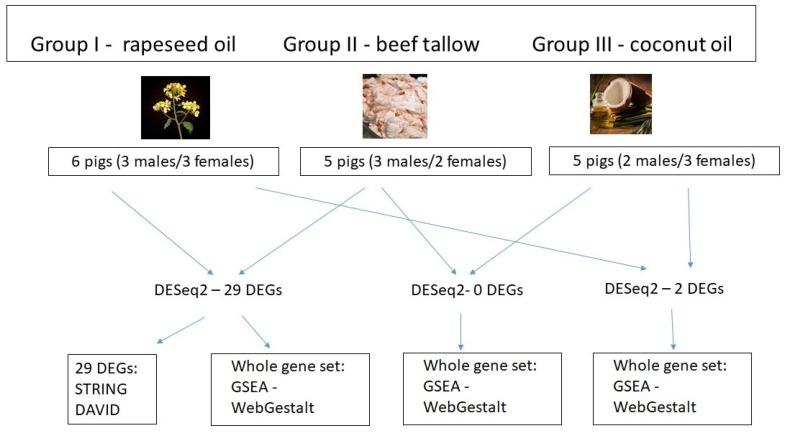
Scheme of the nutritional experiment and functional analyses performed in the study (DEGs: Differentially Expressed Genes, GSEA: Gen Set Enrichment Analysis).

**Figure 2 genes-10-00948-f002:**
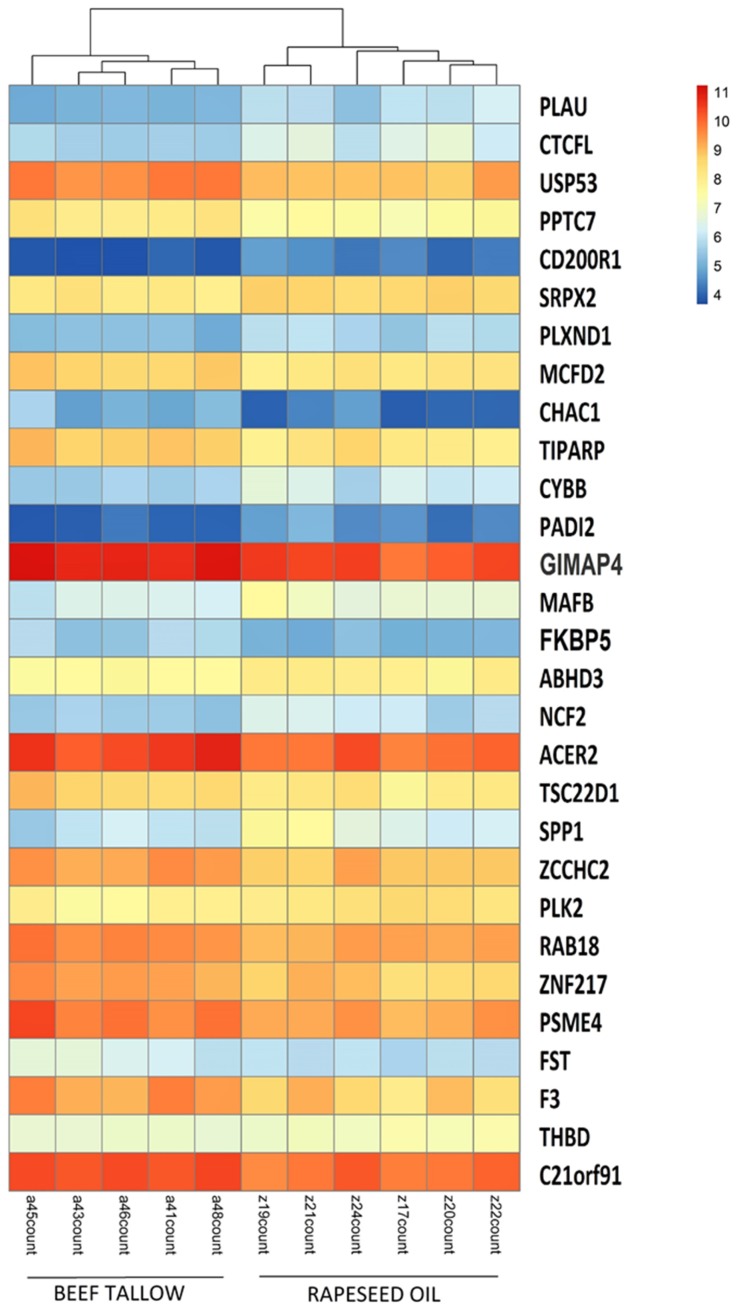
Hierarchical clustering of samples based on the expression level of differentially expressed genes (DEGs).

**Figure 3 genes-10-00948-f003:**
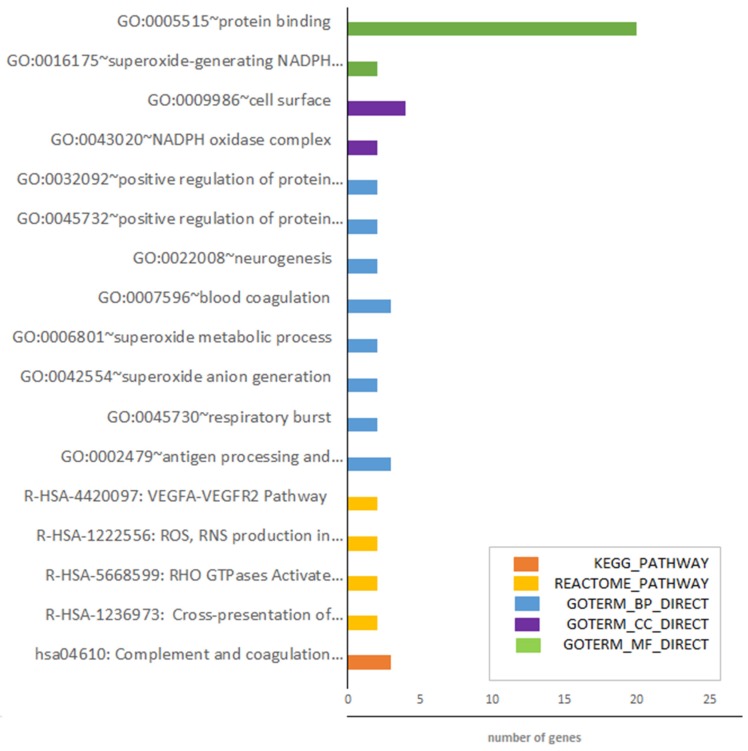
Functional annotation of identified DEGs in rapeseed oil vs. beef tallow comparison.

**Figure 4 genes-10-00948-f004:**
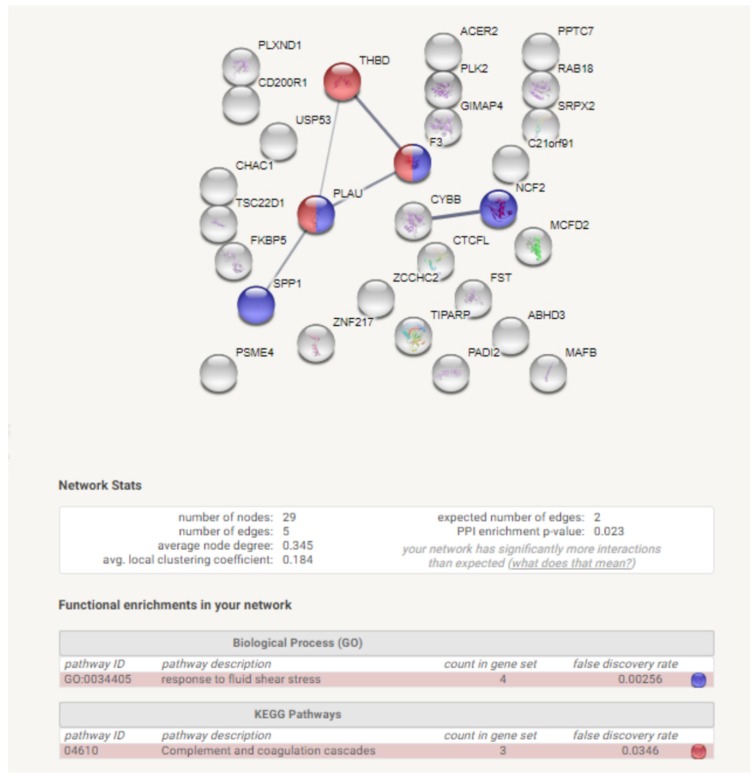
Gene network created from DEGs identified in rapeseed oil vs. beef tallow comparison.

**Figure 5 genes-10-00948-f005:**
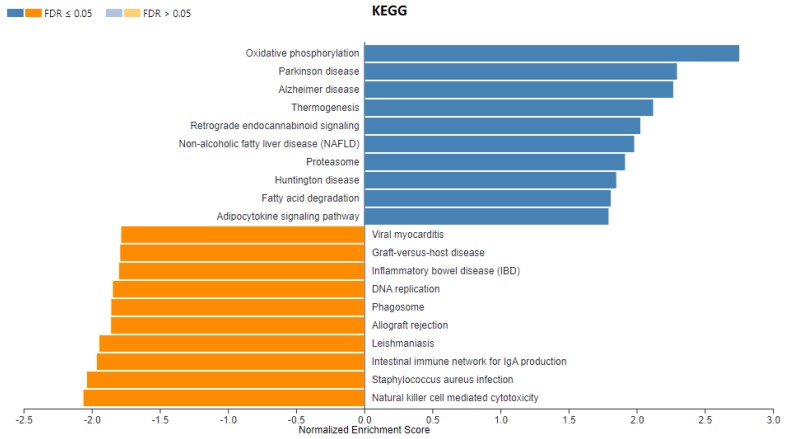
Enriched KEGG pathways *(Sus scrofa)* identified in the rapeseed oil vs. beef tallow dataset by WebGestalt software. Positive normalized enrichment score indicates that genes from the pathway are at the top of the ranked list (mostly upregulated). Negative normalized enrichment score indicates that genes from the pathway are at the bottom of the ranked list (mostly downregulated) (the first group in the comparison is the reference group).

**Figure 6 genes-10-00948-f006:**
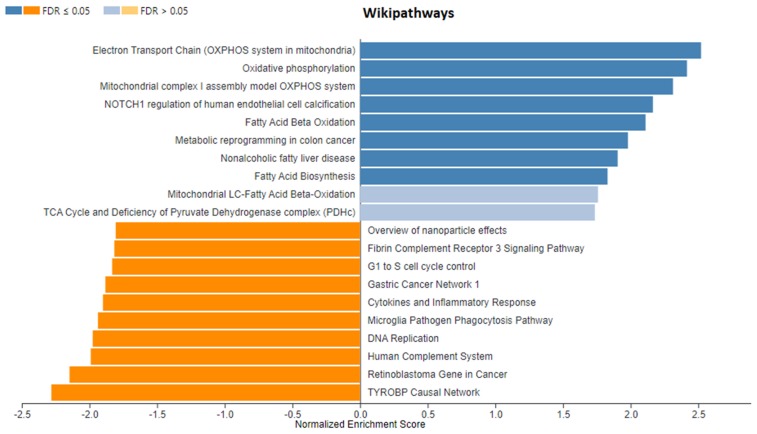
Enriched Wikipathways (Homo sapiens) identified in rapeseed oil vs. beef tallow dataset by WebGestalt software. Positive normalized enrichment score indicates that genes from the pathway are at the top of the ranked list (mostly upregulated). Negative normalized enrichment score indicates that genes from the pathway are at the bottom of the ranked list (mostly downregulated) (the first group in the comparison is the reference group).

**Figure 7 genes-10-00948-f007:**
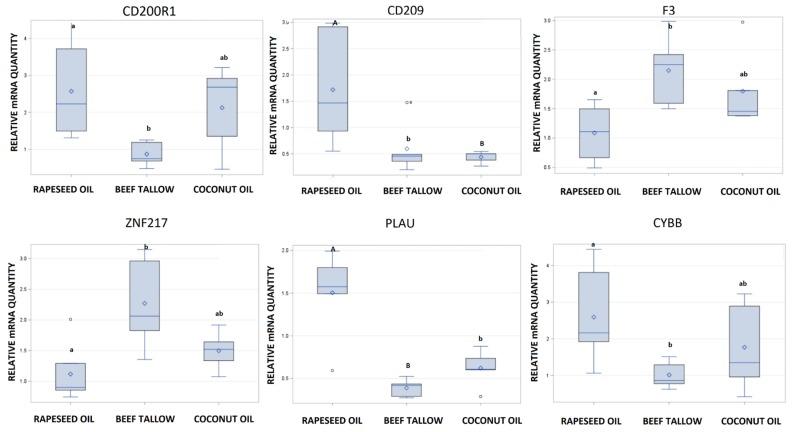
Validation of RNA-seq experiment by qPCR. Error bar-SD. Different letters shows significant differences among groups (small letters: a, b, *p* < 0.05, capital letters: A, B, *p* < 0.01).

**Table 1 genes-10-00948-t001:** Differentially expressed genes identified in the study.

Rapeseed Oil Group vs. Beef Tallow Group				
Ensemble ID	Base Mean	Fold Change	*p* Adjusted	Annotation
ENSSSCG00000010312	50.41389333	−2.3625150	1.13E−06	PLAU
ENSSSCG00000011925	19.57714649	−2.2069404	0.000615734	CD200R1
ENSSSCG00000007505	72.29712452	−2.2061260	5.68E−06	CTCFL
ENSSSCG00000026478	23.05080559	−1.9462864	0.024563058	PADI2
ENSSSCG00000009216	108.6558323	−1.8750910	0.039998048	SPP1
ENSSSCG00000012482	48.83657882	−1.8703811	0.003229498	SRPX2
ENSSSCG00000012229	65.93558552	−1.8667126	0.007760072	CYBB
ENSSSCG00000027826	111.3101373	−1.7931908	0.026147634	GIMAP4
ENSSSCG00000015559	61.23829631	−1.7816443	0.029057113	NCF2
ENSSSCG00000011592	355.9228073	−1.5539315	0.003229498	PLXND1
ENSSSCG00000016925	283.1958901	−1.5317017	0.04197925	PLK2
ENSSSCG00000007115	131.5794279	−1.4363220	0.0489776	THBD
ENSSSCG00000030898	234.0696594	−1.4207480	0.026147634	MAFB
ENSSSCG00000028655	698.3266126	1.4352279	0.04197925	RAB18
ENSSSCG00000021429	1122.072235	1.4361345	0.049411475	C21orf91
ENSSSCG00000004897	580.8230106	1.5350072	0.041653549	ZCCHC2
ENSSSCG00000021325	359.9374829	1.5520128	0.003437732	MCFD2
ENSSSCG00000009422	352.6278443	1.5912329	0.038291764	TSC22D1
ENSSSCG00000007484	548.372453	1.6200279	0.047085346	ZNF217
ENSSSCG00000026955	1269.38782	1.6265332	0.036381897	ACER2
ENSSSCG00000001549	1661.832725	1.6413418	0.026147634	FKBP5
ENSSSCG00000008412	812.4588633	1.6462623	0.047684732	PSME4
ENSSSCG00000016892	71.97654154	1.6559401	0.048396997	FST
ENSSSCG00000022447	568.7564471	1.7298209	0.048396997	F3
ENSSSCG00000027646	386.8914671	1.7299692	0.007003091	TIPARP
ENSSSCG00000028010	42.58112492	1.7664172	0.026147634	ABHD3
ENSSSCG00000009825	239.1954653	1.7791477	2.93E−05	PPTC7
ENSSSCG00000023174	679.5951006	1.8861755	5.68E−06	USP53
ENSSSCG00000004754	30.2851475	2.0652702	0.005616471	CHAC1
Rapeseed Oil Group vs. Coconut Oil Group				
ENSSSCG00000013579	105.0542477	−2.1693396	0.000180283	CD209
ENSSSCG00000020756	25.934304	1.9393441	0.00330069	FAM101A

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
