# Peer review of "Source of Dietary Fat in Pig Diet Affects Adipose Expression of Genes Related to Cancer, Cardiovascular, and Neurodegenerative Diseases"

_genes, 2019, doi:10.3390/genes10120948_

Round 1

Reviewer 1 Report

In this manuscript, the authors reported a small number of genes were differentially expressed between pigs fed with different source of dietary fat. These differentially expressed genes seem interesting since they are related to cancer, cardiovascular and neurodegenerative diseases in humans. Their results imply that dietary animal fat might be implicated in pathogenesis of civilization diseases in human. However, to make their results more convincible and make the manuscript more readable, a couple of questions should be addressed and quite a few writing issues should be corrected. For details, please see the following.

Major issues:

Since the number of differentially expressed genes between different treatment group is very small, instead of gene ontology or pathway overrepresentation analysis, could you perform gene set enrichment analysis using the whole gene list ranked by log2-fold change? See https://www.pnas.org/content/102/43/15545 and https://bmcbioinformatics.biomedcentral.com/articles/10.1186/1471-2105-10-161. GO term overrepresentation analysis based on a very small number of DEGs is not very meaning due to poor statistical power. The gene set enrichment analysis should be more appropriate for this situation, without applying an arbitrary cutoff of significance in advance. Since 16 samples were processed for RNA-seq, could the authors provide details about how the samples were blocked into different batches at each step: RNA preparation, library preparation and library pooling? Did the authors consider the batch effect in the statistical models to identify DEGs? Could the author provide a PCA plot showing sample relationship based the resulting gene expression data? It is not sure why both RSEM and HTSeq software were used for RNA-seq data analysis. Both of them can be used for summarize gene expreesion, but only one is needed for the analysis. Which one was used actually? Since stranded RNA-seq was performed, did the authors make use of the strandedness information in the analysis process? If so, how? Line 128, what were the models for DEG analysis using DESeq2? Lines 134-135, how did the authors use both GLM (parametric) and Mann-Whitney test (non-parametric) for qPCR data analysis?

Minor issues:

Could you explicitly describe the number of pigs in each treatment group and the criteria used to define DEGs in Abstract? Lines 16-17, the sentence is not appropriate. “in response to feeding pigs with different source of fat”? Line 61, pleas double check the sentence: “…none of the dietary factor s (cDDGS or fat) did not affect …”. Line 64, “deeper investigate”? Line 66, “insight” or “insights”? Line 68, remove the first dot. Line 74, could you give more details about the sampling sites of the adipose tissues used for RNA-seq? Line 88, could you provide more details about pig slaughtering? How were pigs anesthetized and euthanized? Lines 92-93, change all “DDGS”s to “cDDGS”s. Line 112, “synthesized”? How much total RNA was used for each RT reaction? Line 126 and other places, could you please list the version of software used for the analyses? How were major parameters set? What is the version of GTF file used for genome annotation? Line 128, what method was used for p-value adjustment? “fold-change > ±3” is not reasonable. Fold change is always great than 0. Did the author mean log2-fold change? What was the quality of the library in terms of strandedness? See the report from the RSeQC report. Lines 139, the authors report more than 82% of the reads were mapped to the reference genome. The mapping rates were low. Usually, more than 90% of reads are uniquely mapped, ~5% multi-mapper. Could you explain why? All description of the results of DEG analysis is not very clear. When mentioned up- and down-regulated genes, please explicitly specify the compared groups and which the reference group is. Figure 1. What is the unit of the heatmap key? Count or TPM or something else? For heatmap, usually standardized expression values (Z-scores) are used. Table 1 and Supplementary Table 1. Please change the decimal symbol from “,” to “.”. Lines 179, “R2” should be “R^2”. Figure 4 and Lines 111-121, how many RNA samples were used for qPCR validation? Were they the same samples used for RNA-seq? How many technical replicates were used for qPCR? What’s the error bar, SD or SE? What are the letters above each bar? The axis labels were barely readable. Could you use better quality plots? Could you show the correlation between RNA-seq and qPCR by using a scatter plot? Line 197, “String” or “STRING”. Lines 221-222, please rephrase the sentence. Line 294, move “P < 0.055” to the end of the sentence and inside parentheses. Line 327, remove “[“. Line 344, change the comma to a period. In References, all the journal names should be italic. See Line 421, 425, and so on.

Author Response

1. Since the number of differentially expressed genes between different treatment group is very small, instead of gene ontology or pathway overrepresentation analysis, could you perform gene set enrichment analysis using the whole gene list ranked by log2-fold change? See https://www.pnas.org/content/102/43/15545 and https://bmcbioinformatics.biomedcentral.com/articles/10.1186/1471-2105-10-161. GO term overrepresentation analysis based on a very small number of DEGs is not very meaning due to poor statistical power. The gene set enrichment analysis should be more appropriate for this situation, without applying an arbitrary cutoff of significance in advance.

According to the reviewer`s suggestions we performed GSEA using WEB Gestalt software (see results and discussion sections)

2. Since 16 samples were processed for RNA-seq, could the authors provide details about how the samples were blocked into different batches at each step: RNA preparation, library preparation and library pooling? Did the authors consider the batch effect in the statistical models to identify DEGs? 

The information was added to the text (lines 109-111)

3. Could the author provide a PCA plot showing sample relationship based the resulting gene expression data?

PCA plots have been added to the supplementary materials 

4. It is not sure why both RSEM and HTSeq software were used for RNA-seq data analysis. Both of them can be used for summarize gene expreesion, but only one is needed for the analysis. Which one was used actually?

HTSeq was used, RSEM was deleted from the text

5.Since stranded RNA-seq was performed, did the authors make use of the strandedness information in the analysis process? If so, how

6. what were the models for DEG analysis using DESeq2?

7.how did the authors use both GLM (parametric) and Mann-Whitney test (non-parametric) for qPCR data analysis?

We used Mann-Whitney test only (corrected)

8. Could you explicitly describe the number of pigs in each treatment group and the criteria used to define DEGs in Abstract?

The information was added to the abstract

9. Lines 16-17, the sentence is not appropriate. “in response to feeding pigs with different source of fat”? 

The sentence was corrected

10. Line 61, pleas double check the sentence: “…none of the dietary factor s (cDDGS or fat) did not affect …”. 

The sentence was corrected

11.  Line 64, “deeper investigate”? 

The sentence was corrected

12. Line 66, “insight” or “insights”?

I think insight is ok

13.Line 68, remove the first dot. 

corrected

14. Line 74, could you give more details about the sampling sites of the adipose tissues used for RNA-seq? 

The samples were collected from the area between the last thoracic and the first lumbar vertebrae.  - the information was added to the text

16. Line 88, could you provide more details about pig slaughtering? How were pigs anesthetized and euthanized?

The pigs were slaughtered by stunning with high-voltage electric tongs (voltage 240–400 V) - information was added to the text (lines 89-90)

17.Lines 92-93, change all “DDGS”s to “cDDGS”s. 

corrected

18.  Line 112, “synthesized”

corrected

19. How much total RNA was used for each RT reaction?

cDNA was synthesized from 500 ng of total RNA - the information was added to the text (line117)

20 Line 126 and other places, could you please list the version of software used for the analyses?

21. How were major parameters set?

22. What is the version of GTF file used for genome annotation?

23. Line 128, what method was used for p-value adjustment?

“fold-change > ±3” is not reasonable. Fold change is always great than 0. Did the author mean log2-fold change?

25. What was the quality of the library in terms of strandedness?

26. Lines 139, the authors report more than 82% of the reads were mapped to the reference genome. The mapping rates were low. Usually, more than 90% of reads are uniquely mapped, ~5% multi-mapper. Could you explain why?

The reason of low mapping rates is that  Sus scrofa genome is not as well annotated as human`s or rat`s genomes. 

 27. All description of the results of DEG analysis is not very clear. When mentioned up- and down-regulated genes, please explicitly specify the compared groups and which the reference group is. 

We added to the text additional  explanation of comparisons: (rapeseed oil vs beef tallow, rapeseed vs. coconut oil, beef tallow vs coconut oil, the first group in the comparison is the reference group) - line 149-150

Reviewer 2 Report

Oczkowicz et al. addressed an important issue to demonstrate the transcriptomic data expression in pig animal model under dietary fats consumption. The articles are comprehensive and well-written. However, I still have some comment for this manuscript.

1. How to calculate the p-adjusted value in this study? Please clarify the statistical part. If multiple comparison, multiple testing should be considered.

2. Since the study investigate the gene expression in adipose tissue of pigs, is the adipose tissue from different resource? There are white fat and brown fat adipose tissue. Which part does this experiment perform and compare?

3. How to select six differentially expressed genes (CD200R1, CD209, CYBB, ZNF217, F3, PLAU) for qPCR analysis validation in this study? Please clarify. How about other gene expression difference in beef tallow group vs rapeseed oil group? Would the author consider other pathway analysis tool, such as Ingenuity Pathway Analysis, and also select the key gene regulation in the transcriptome data?

4.The author could provide a study flowchart to present the experiment and data analysis process.

5.The landmark in the Figure 4 is not very clear. Please revise the figure.

Author Response

Dear reviewer,

Thank you very much for your suggestions.

Maria Oczkowicz

How to calculate the p-adjusted value in this study? Please clarify the statistical part. If multiple comparison, multiple testing should be considere

FDR - Benjamini-Hochberg (BH) adjustment was used – line136

2.Since the study investigate the gene expression in adipose tissue of pigs, is the adipose tissue from different resource? There are white fat and brown fat adipose tissue. Which part does this experiment perform and compare?

 The experiment regards white subcutaneous adipose tissue. The information was added in Line 87

How to select six differentially expressed genes (CD200R1, CD209, CYBB, ZNF217, F3, PLAU) for qPCR analysis validation in this study? Please clarify. How about other gene expression difference in beef tallow group vs rapeseed oil group? Would the author consider other pathway analysis tool, such as Ingenuity Pathway Analysis, and also select the key gene regulation in the transcriptome data?

Differential expressed genes for QPCR analysis were selected based on fold change ( we selected genes with high fold change: PLAU, CD200R1 and with low fold change: ZNF217, F3). The second criterion was the availability of TaqMan assays. We agree that IPA analysis could provide important information to our experiment, unfortunately we do not have a license to run this analysis. Instead, we used free WEB Gestalt software to perform Gene Set Enrichment Analysis as suggested by the second reviewer.

4.The author could provide a study flowchart to present the experiment and data analysis process.

The study flowchart has been provided as Figure 1

5.The landmark in the Figure 4 is not very clear. Please revise the figure.

The Figure has been revised

Round 2

Reviewer 1 Report

The revised version is significantly improved. But there are still quite a few minor issues. please carefully proof read the whole manuscript

Lines 26 and 602, remove the “:” after  “both”.

Line 44, delete “an” before “insights”

Line 425, remove the  “:”.

Line 459, missing closing parenthesis.

Supplementary Figure 2, PCA analysis is better done with the whole dataset considered, instead of two groups of samples separately. Do you perform PCA using only the normalized expression data of the DEGs?

Supplementary Table 2,  for decimals, using "." instead of ",".

Author Response

Lines 26 and 602, remove the “:” after  “both”.

corrected

Line 44, delete “an” before “insights”

corrected

Line 425, remove the  “:”.

corrected

Line 459, missing closing parenthesis.

corrected

Supplementary Figure 2, PCA analysis is better done with the whole dataset considered, instead of two groups of samples separately. Do you perform PCA using only the normalized expression data of the DEGs?

Yes, we did PCA analysis using only the normalized expression data of the DEGs (rld method)

Supplementary Table 2,  for decimals, using "." instead of ",".

corrected

Reviewer 2 Report

All comments and suggestions had been addressed. I have no more question.

Author Response

Thank you!